# Bending Strength Design Method of *Phyllostachys edulis* Bamboo Based on Classification

**DOI:** 10.3390/polym14071418

**Published:** 2022-03-30

**Authors:** Pengcheng Liu, Qishi Zhou, Feiyang Fu, Wei Li

**Affiliations:** School of Civil Engineering, Central South University, Changsha 410075, China; liupengcheng@csu.edu.cn (P.L.); 204812376@csu.edu.cn (F.F.); 204812291@csu.edu.cn (W.L.)

**Keywords:** *P. edulis* bamboo, grade, bending test, reliability analysis

## Abstract

*Phyllostachys edulis* (*P. edulis*) bamboo is the most widely distributed and used bamboo species, and it is an ideal building material. With the in-depth implementation of the sustainable development strategy, modern bamboo structures have broad application prospects in green buildings. In order to promote the efficient utilization of bamboo resources and facilitate the design and application of bamboo structures, the bending strength test and classification of *P. edulis* bamboo were carried out, the factors affecting the reliability were analyzed, and the design values of the bending strength of *P. edulis* bamboo were proposed based on the reliability analysis. The research results show that dividing *P. edulis* bamboo into three levels (grade I, grade II, and grade III) can achieve efficient use of *P. edulis* bamboo resources; 75% fitting data points and normal distribution were used to analyze the reliability of the bending strength of *P. edulis* bamboo. The analysis of factors affecting reliability makes the calculation of strength design values more reliable. The reliability increases with the increase of the load ratio and the partial factor for resistance. Under the same load ratio and reliability, the partial factor for resistance of the combination of constant load and snow load is the largest, and the partial factor for resistance of the combination of constant load and office building load is the smallest. Under the same load combination and reliability, the partial factor for resistance decreases as the load ratio increases. Under the same load ratio and load combination, the partial factor for resistance of grade III is the largest, and grade I is the smallest. The bending strength design values of grade I, grade II, and grade III are 29.54 MPa, 29.62 MPa, and 30.63 MPa, respectively. This paper innovatively proposed the design values of bending strength of *P. edulis* bamboo based on classification. The *P. edulis* bamboo grading method established in this paper and the bending strength design values of *P. edulis* bamboo proposed can provide references for the design and engineering application of bamboo structures.

## 1. Introduction

There are abundant bamboo resources in the world, mainly distributed in the Asia-Pacific region, the Americas, and Africa. China is one of the countries with the widest distribution, largest resources and earliest utilization of bamboo in the world, and is known as the “Kingdom of bamboo”. In addition to being a crop, bamboo is also often used in many aspects such as construction and life. As a building material, bamboo has the following advantages [1,2,3,4,5]: (1) Bamboo is the fastest growing plant on earth. (2) Bamboo is one of the hardest plants in the world, and its strength varies with bamboo age, location, and type. (3) Bamboo is soft and elegant in color, clear and delicate in texture, giving people a double enjoyment of vision and psychology, which is incomparable to other materials. (4) Compared with other building materials, bamboo is simple in design and flexible in construction, and they can be regularly maintained by replacing damaged parts. (5) Due to its light weight and good elasticity, the seismic function of bamboo is very outstanding. (6) Bamboo buildings are beautiful in appearance and high in comfort. (7) Bamboo is easy to grow in many countries in the world. (8) The processed bamboo has the characteristic that it can keep its original performance unchanged even under the influence of long-term general damage. Therefore, bamboo is an ideal building material. Since 2013, bamboo structure and bamboo engineering materials have been listed as green buildings and green materials in a series of policy documents in China, including the Action Plan on Green Building, Action Plan on Promoting the Production and Application of Green Building Materials, and The National New Urbanization Plan (2014–2020).

A series of achievements have been made in the research of bamboo in the field of construction. Bamboo is an anisotropic material with strong tensile, compressive and bending properties [6,7,8,9,10]. The mechanical properties of bamboo are related to many factors such as height, wall thickness, diameter, density, and moisture content [11,12,13,14]. The mechanical properties of bamboo along the grain direction increase with the increase of height [11] and decrease with the increase of wall thickness and diameter [12]. The mechanical properties of bamboo along the grain direction are positively correlated with density [13], and negatively correlated with moisture content [14]. The prediction of the mechanical properties of bamboo can be achieved through the fitting relationship between the mechanical properties of bamboo and the height, wall thickness, diameter, density, and moisture content. The nodes affect the mechanical properties of bamboo. Generally speaking, the mechanical properties of node specimens are higher than that of internode specimens [15]. Tian et al. [16] carried out theoretical and finite element analysis on bending of bamboo beams, and the results show that bamboo beams have good bending performance and proposed a load-deflection curve calculation method. Bahtiar et al. [17] conducted an experimental study on the buckling performance of bamboo stalks of different lengths, and the results show that bamboo columns have good compressive performance and proposed relevant buckling formulas. Tian et al. [18] proposed a composite slab with thermal insulation material sprayed outside the original bamboo skeleton. Considering the composite effect of bamboo and thermal insulation material, the bending capacity and stiffness of composite slab were obtained. The research results show that the composite effect of bamboo and insulation materials is good. Hu et al. [19] studied a bamboo tube joint connected by a steel hoop, analyzed the friction resistance of the steel hoop connection, and proposed a calculation method for the bearing capacity of the interface between the bamboo tube and the steel hoop. Although the research of experts and scholars on the mechanical properties and components of bamboo has confirmed the feasibility of bamboo in the promotion and application of construction, the design and application of bamboo are restricted due to the lack of strength design values. The strength design values are the basis of bamboo structure design and application. Before studying the strength design values, it is necessary to classify the bamboo. Classification can realize the efficient use of bamboo resources. At present, the allowable stress method and limit state design method are the main methods used for strength design value analysis, among which the limit state design method is internationally recognized [20,21,22]. Based on the limit state design method of reliability, Liu [20] studied the strength design values of bamboo along the grain compressive strength, along the grain tensile strength, transverse grain compressive strength and transverse grain tensile strength. Zhu [21] proposed a theory to determine the strength design values of wood based on the above methods. Due to the mature application of limit state design method, this paper also uses this method to analyze the design values of bending strength of bamboo.

In order to classify the *P. edulis* bamboo and propose the design values of bending strength, the following work was carried out in this paper: (1) The bending test of bamboo was carried out, and the failure process of bamboo bending was analyzed. (2) The factors affecting reliability were analyzed. (3) The bending partial factors for resistance and strength design values of bamboo were determined.

## 2. Materials and Methods

### 2.1. Materials

The producing areas of the *P. edulis* bamboo studied in this paper are Hunan and Zhejiang, China, each accounting for half of them. Firstly, bamboos with an age of about 4 years and no obvious damage were randomly selected from the bamboo forest for felling (Figure 1a), and then transported back to the laboratory. After the air-drying treatment, the bending specimens were prepared (Figure 1b). The sampling location of the specimens ranged from 0 to 6 m. The bending specimen size is 220 mm × 15 mm × *t* mm (*t* is the wall thickness).

### 2.2. Bending Test

The four-point method was used for loading (Figure 1c). The two loading points are each 50 mm away from the center line of the support, and the distance between the loading points is 80 mm ± 1 mm. The specimens were placed on two supports of the test apparatus. Three dial indicators are installed on the base of the testing machine to measure the deflection of the two loading points and the middle of the span. The test loading rate was 150 N/mm^2^ per minute. The calculation formulas of bending strength and elastic modulus are as [23]:(1)$$MOR_{W}=\frac{150P_{\max}}{tb^{2}}$$
(2)$$MOE_{W}=\frac{1,920,000\Delta P}{8\delta_{m}tb^{3}}$$
where *MOR*_W_ is bending strength under water content W (MPa); *MOE*_W_ is the modulus of flexural elasticity under water content W (MPa); *P*_max_ is failure load (N); *A* is the stressed area (mm^2^); *t* is specimen thickness (mm); *b* is the height of specimen (mm); ∆*P* is the difference between upper and lower loads (N); *δ*_m_ is the deflection value of pure bending section of specimen under ∆*P* action (mm).

After the specimen was damaged, a small specimen with a size of about 20 mm × 20 mm × *t* mm was cut near the damaged location to measure the moisture content. The calculation formula of moisture content is as follows [24]:(3)$$W=\frac{m_{1}-m_{0}}{m_{0}}\times 100$$
where *W* is air-dry moisture content (%); *m*_1_ and *m*_0_ are the mass of air drying and full drying, respectively (g).

The bending strength and elastic modulus were adjusted to the values under standard moisture content (12%), and the adjustment formulas are as follows [23]:

*MOR*_12_ = *K*_MOR_*MOR*_W_(4)

*MOE*_12_ = *K*_MOE_*MOE*_W_(5)

(6)$$K_{\mathrm{MOR}}=\frac{1}{0.971+0.217e^{-0.2w}}$$(7)$$K_{\mathrm{MOE}}=\frac{1}{0.91+0.3e^{-0.1w}}$$
where *MOR*_12_ is the bending strength under standard moisture content (MPa); *MOE*_12_ is the bending elastic modulus under standard moisture content (GPa); *K*_MOR_ and *K*_MOE_ are the moisture content correction coefficients of the bending strength and elastic modulus.

### 2.3. Reliability Analysis

At present, there are many reliability analysis methods; the central point method is one of the most commonly used. According to the central point method, the reliability calculation formula is obtained as shown in Formula (8) [24]. The advantage of Formula (8) is that it gives the calculation method of reliability directly, which greatly simplifies the calculation.
(8)$$\beta =\frac{K_{R}\gamma_{R}(\gamma_{G}+\gamma_{Q}\rho )-(K_{G}+K_{Q}\rho )}{\sqrt{{\left[K_{R}\gamma_{R}(\gamma_{G}+\gamma_{Q}\rho )\delta_{R}\right]}^{2}+{\left(K_{G}\delta_{G}\right)}^{2}+{\left(K_{Q}\rho \delta_{Q}\right)}^{2}}}$$
where *K*_R_ is the uncertainty of resistance, *δ*_R_ is the coefficient of resistance variation; *γ*_R_ is the partial coefficient of resistance, *γ*_G_ is the permanent load component coefficients, *γ*_Q_ is the variable load component coefficients; *K*_G_ is the permanent load uncertainty coefficients, *δ*_G_ is the variation coefficient, *K*_Q_ is the variable load uncertainty coefficient, *δ*_Q_ is the variation coefficients; *ρ* is the ratio of the variable load to the standard value of the permanent load effect.

## 3. Results and Discussion

### 3.1. Destruction Process and Destruction Form

The bending failure process of bamboo is as follows: at the initial stage of loading, the specimen is in an elastic state, and the deflection of the specimen increases linearly with the increase of the load. With the further increase of the load, the deflection of the specimen increases nonlinearly with the increase of the load, and the specimen has a deflection visible to the naked eye. When the fracture occurs in the compression zone at the loading point, the specimen is destroyed, as shown in Figure 1d. Typical load-displacement curves of specimens are shown in Figure 2. Figure 2 shows that the specimen had no obvious platform segment. There was no obvious sign before the failure of the specimens, so the bending failure mode of bamboo was a brittle failure.

### 3.2. Classification

Bending elastic modulus is usually used as the classification index of materials. The distribution of *MOE* is shown in Figure 3. It can be seen from Figure 3 that the distribution of *MOE* conforms to the characteristics of the normal distribution, and the data is mainly concentrated between 16.5 GPa and 18.5 GPa. Therefore, based on the stability of *MOE* distribution, *MOE* can be used as an index to evaluate mechanical properties in the performance evaluation system of *P. edulis* bamboo. Based on the above analysis, the bending elastic modulus of moso bamboo is divided into three grades (grade I, grade II, and grade III), and the classification results are shown in Table 1. The range less than 16.5 GPa is regarded as grade I, the range not less than 16.5 GPa and less than 18.5 GPa is regarded as grade II, and the range larger than 18.5 GPa is classified as grade III. The mean values of *MOE* of grade I, grade II, and grade III are 15.90 GPa, 17.41 GPa, and 19.22 GPa, respectively. Grade II is 9.48% higher than grade I, and grade III is 10.43% higher than grade II. The proportions of grade I, grade II, and grade III are 14.19, 65.16, and 20.65%, respectively, of which the sum of the proportions of grade II and grade III exceeds 80%. Literature [25] also adopted the above method for classification. Through the classification of grades, the efficient use of *P. edulis* bamboo resources can be realized.

### 3.3. Strength Statistical Analysis

Normal distribution, lognormal distribution, and Weibull distribution are usually used to fit the distribution of material strength. Table 2 shows the statistical results of bamboo bending strength. It can be seen from Table 2 that there are differences in bending strength under different grades. From grade I to grade III, the mean bending strength increases gradually. The average bending strengths of grade I, grade II, and grade III under normal distribution are 130.94 MPa, 131.84 MPa, and 136.46 MPa, respectively. Grade II is 0.69% higher than grade I, and grade III is 3.5% higher than grade II. The coefficient of variation of the bending strength of different grades is also different, the highest is 0.060, and the lowest is 0.047. There are also differences in the results obtained by using different distribution types to fit the bending strength. The mean value of bending strength fitted by the lognormal distribution is the highest, followed by the normal distribution, and the Weibull distribution is the smallest. The coefficient of variation of bending strength obtained by Weibull distribution fitting is the largest. Generally, the 5% quantile value under the 75% confidence level is used as the strength standard value. The standard values of the bending strength of *P. edulis* bamboo at each grade are shown in Table 2. The standard value of strength decreases with the decrease of fitting data points and increases with the increase of grade.

### 3.4. Analysis of Factors Affecting Reliability

Probability distribution types and fitting data points are important factors affecting reliability [26]. In order to analyze the influence of fitting data points and probability distribution types on reliability, the bending strength was ranked from small to large, and then the strength values were divided according to the number of different fitting data points. The number of fitting data points considered in this paper is the first 100% (total is *N*), the first 75%, the first 50%, and the first 25%, respectively, and the probability distribution types considered include normal distribution, lognormal distribution and Weibull distribution. Statistical results of bending strength under different fitting data points and probability distribution types are shown in Table 2. As can be seen from Table 2, with the decrease of fitting data points, the mean value of bending strength and coefficient of variation under different probability distributions both decrease.

The tail cumulative probability density has an important influence on reliability [26]. Figure 4 shows the fitting results of the cumulative probability of bending strength of grade II bamboo under different probability distributions. As can be seen from Figure 4, when the cumulative probability density is lower than 0.25, the curve obtained by fitting normal distribution and lognormal distribution is relatively close, and the curve obtained by fitting Weibull distribution is higher than that obtained by normal distribution and lognormal distribution. The difference of curves obtained by fitting 100% data points with three distribution models is the largest, and the difference of 75, 50, and 25% data points is reduced.

The relationship between *β* and *γ*_R_ was obtained by reliability analysis as shown in Figure 5 (only show constant load + live load of the residential floor). As can be seen from Figure 5, *β* increases gradually with the increase of *γ*_R_. The curves obtained by using normal distribution and lognormal distribution almost coincide, while the curves obtained by using Weibull distribution are different from those obtained by using normal distribution and lognormal distribution to some extent. Curves of the three distributions with 100% data points have the greatest difference. With the decrease of fitting data points, the difference of curves gradually decreases. For ductile failure, the target reliability *β*_0_ is 3.2, and for brittle failure, the target reliability *β*_0_ is 3.7 [27]. In the range where *β* is lower than 3.7, the proximity of the curves under the three distributions is higher and almost identical at 75, 50, and 25% data points. In order to involve as many data as possible in the analysis, this paper selects 75% data points for research.

In order to further determine the type of probability distribution, this paper analyzes P–P graphs of different distributions under 75% data points, and Figure 6 is a P–P graph of grade II. Figure 6 shows that Weibull distribution has the lowest fitting degree, and normal distribution and lognormal distribution have similar fitting effects. In this paper, normal distribution is used for reliability analysis in the next step.

### 3.5. Determination of Partial Factors for Resistance

In reliability analysis, four load combinations are considered, which are constant load + live load of residential floor (D + R), constant load + live load of office floor (D + O), constant load + wind load (D + W) and constant load + snow load (D + S). The return periods of wind load and snow load are 30 years and 50 years, respectively [27,28]. Seven load ratios (ratio of variable load to permanent load) are considered, which are 0, 0.25, 0.5, 1.0, 2.0, 3.0 and 4.0 respectively. The structural safety grade shall be considered as level 2, and the design base year shall be considered as 50 years. Figure 7 is the relationship curve between *β* and *γ*_R_ of grade II data under different load types and load ratios, and Figure 8 is the relationship curve between *β* and *ρ* of grade II data under different load types and partial factors for resistance. It can be seen from Figure 7 that *β* increases with the increase of *γ*_R_, and the slope of the curve gradually decreases. It can be seen from Figure 8 that *β* increases with the increase of *ρ*, and the slope of the curve gradually slows down with the increase of *ρ*. The above analysis results are consistent with the literature [22,26,29]. According to Formula (8), when *ρ* and other parameters are constant, the relationship between *β* and *γ*_R_ is positively correlated with slope decline, which is consistent with the results shown in the curve. When *γ*_R_ and other parameters are constant, the relationship between *β* and *ρ* is also positively correlated with slope decline, which is consistent with the results in Figure 8.

The partial factors for resistance under the target reliability (*β*_0_ = 3.7) are shown in Table 3. It can be seen from Table 3 that when the load ratio is 0, the partial factors for resistance under the four load combinations are equal. Under the same grade and load combination, as the load ratio increases, the partial factor for resistance gradually decreases. Under the same load ratio, the order of partial factor for resistance is: D + S > D + W > D + R > D + O. Under the same load ratio and load combination, the order of partial factor for resistance is: Grade III > Grade II > Grade I. In order to ensure the safety of the structure, this paper takes the largest partial factors for resistance as the indexes for calculating the strength design values of each grade.

### 3.6. Determination of Design Values of Bending Strength

Based on the coefficient of resistance determined by the above analysis, the strength design value of bamboo can be calculated. The formula for calculating the strength design value of *P. edulis* bamboo is as follows [30]:(9)$$f_{d}=\frac{f_{k}K_{P}K_{A}K_{Q}}{\gamma_{R}}$$
where *f*_d_ is the designed strength value of *P. edulis* bamboo (MPa); *f*_k_ is the standard strength value of *P. edulis* bamboo (MPa). *K*_P_ is the calculation model uncertainty coefficient; *K*_A_ is the number of geometric parameter uncertainties; *K*_Q_ refers to the conversion coefficient of small specimen into component [30]. The bending strength design values of *P. edulis* bamboo of all grades are calculated by Formula (9), as shown in Table 4.

## 4. Conclusions

This paper carried out the bending test of the *P. edulis* bamboo, classified the *P. edulis* bamboo, and analyzed the strength design value of the *P. edulis* bamboo. The main conclusions are as follows:*MOE* was used to classify bamboo grades. The grades and corresponding *MOE* ranges were Grade I (*MOE* < 16.5 GPa), Grade II (16.5 GPa ≤ *MOE* < 18.5 GPa), and Grade III (*MOE* ≥ 18.5 GPa).With the decrease of fitted data points, the cumulative probability density curves and reliability-partial factor for resistance curves under normal distribution, lognormal distribution, and Weibull distribution become closer; 75% data points and normal distribution were used to analyze the reliability of *P. edulis* bamboo.With the increase of partial factor for resistance and load ratio. The partial factor for resistance under target reliability decreases with the increase of load ratio. The partial factor for resistance under constant load and snow load combination is the largest, and the partial factor for resistance under constant load and office building load combination is the smallest. Under the same load ratio and load combination, the partial factor for resistance of grade III is the largest, while that of grade I is the smallest.The bending strength design values of *P. edulis* bamboo are as follows: grade I (29.54 MPa), grade II (29.62 MPa), grade III (30.63 MPa). The classification method and the bending design values of *P. edulis* bamboo presented in this paper can provide a reference for the design and engineering application of bamboo structure.

## Figures and Tables

**Figure 1 polymers-14-01418-f001:**
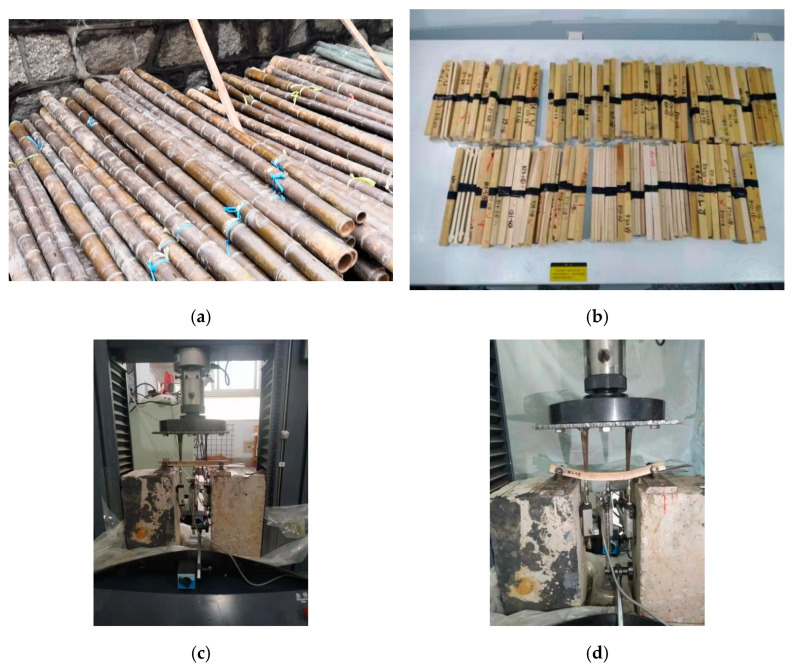
Test and destruction of *P. edulis* bamboo: (**a**) bamboo stalks; (**b**) specimens; (**c**) loaing; (**d**) specimen damage.

**Figure 2 polymers-14-01418-f002:**
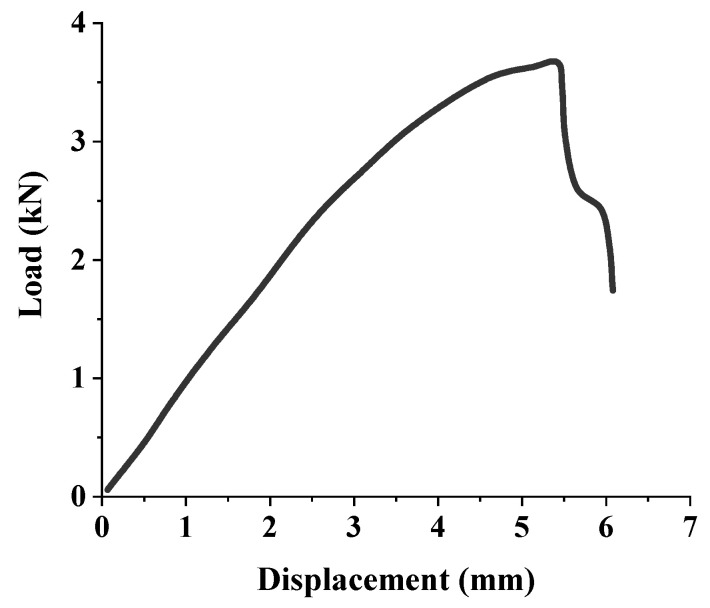
Typical load-displacement curves.

**Figure 3 polymers-14-01418-f003:**
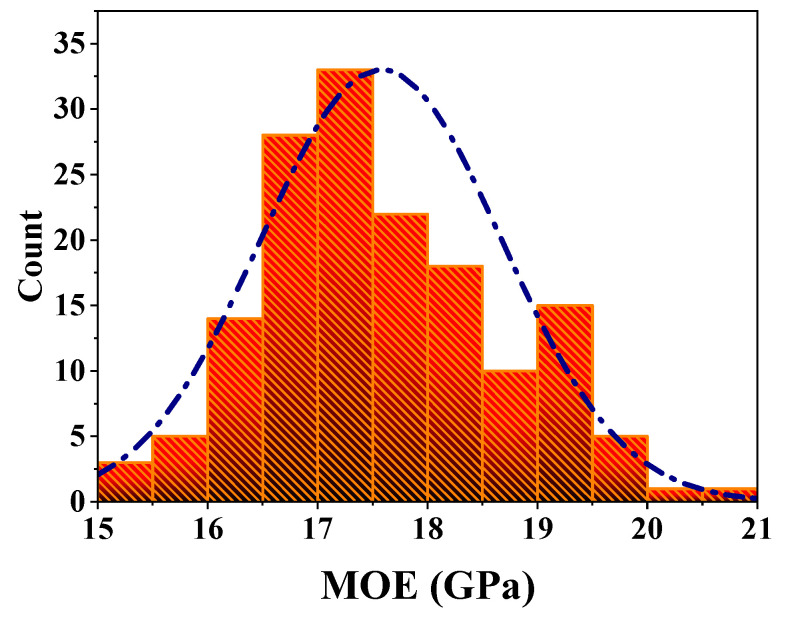
Distribution of *MOE*.

**Figure 4 polymers-14-01418-f004:**
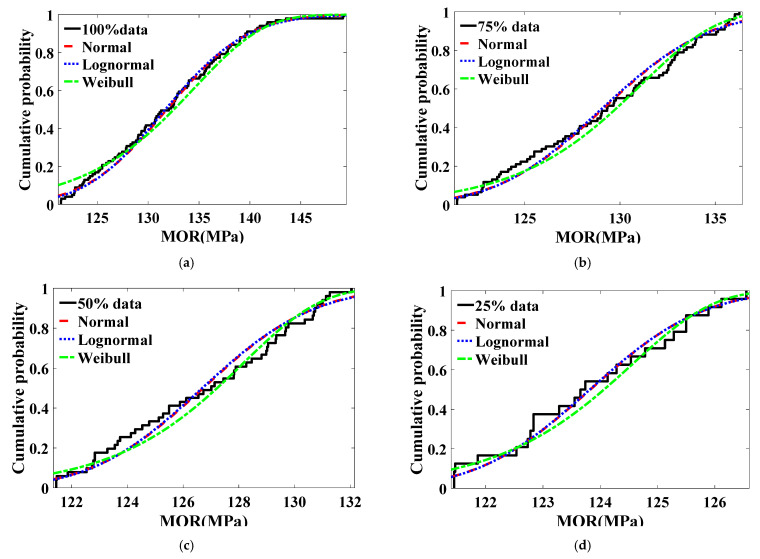
Cumulative probability fitting results of bamboo bending strength under different probability distributions (grade II): (**a**) *n* = 100%N; (**b**) *n* = 75%N; (**c**) *n* = 50%N; (**d**) *n* = 25%N.

**Figure 5 polymers-14-01418-f005:**
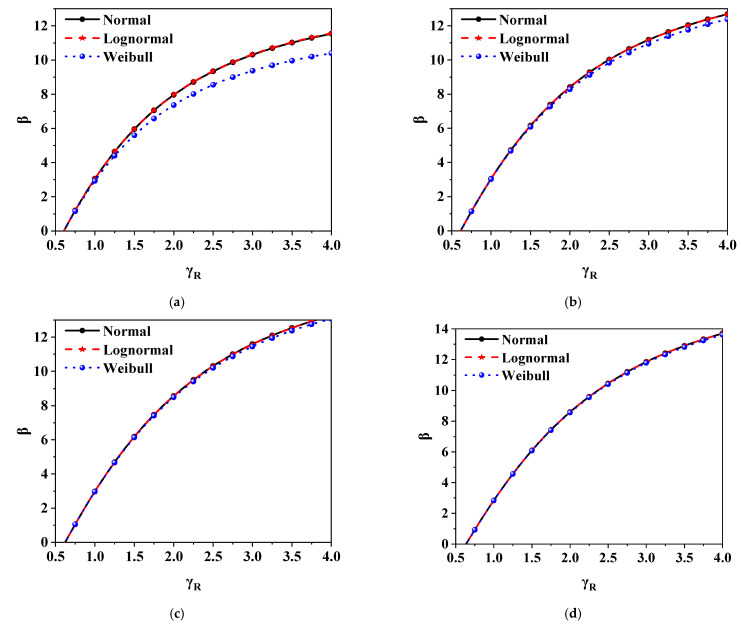
The relationship between *β* and *γ*_R_ under different fitted data points and probability distribution types (grade II, constant load + live load of residential floor, *ρ* = 1.0): (**a**) *n* = 100%N; (**b**) *n* = 75%N; (**c**) *n* = 50%N; (**d**) *n* = 25%N.

**Figure 6 polymers-14-01418-f006:**
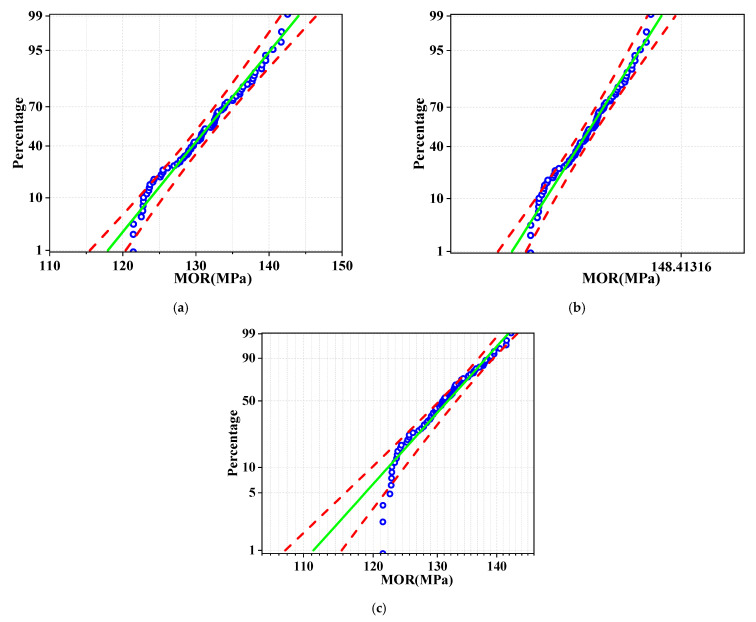
P–P diagram of 75% data under different probability distribution types (grade II): (**a**) normal; (**b**) lognormal; (**c**) Weibull. (blue: Weibull; red: Normal distribution; green: Lognormal distribution).

**Figure 7 polymers-14-01418-f007:**
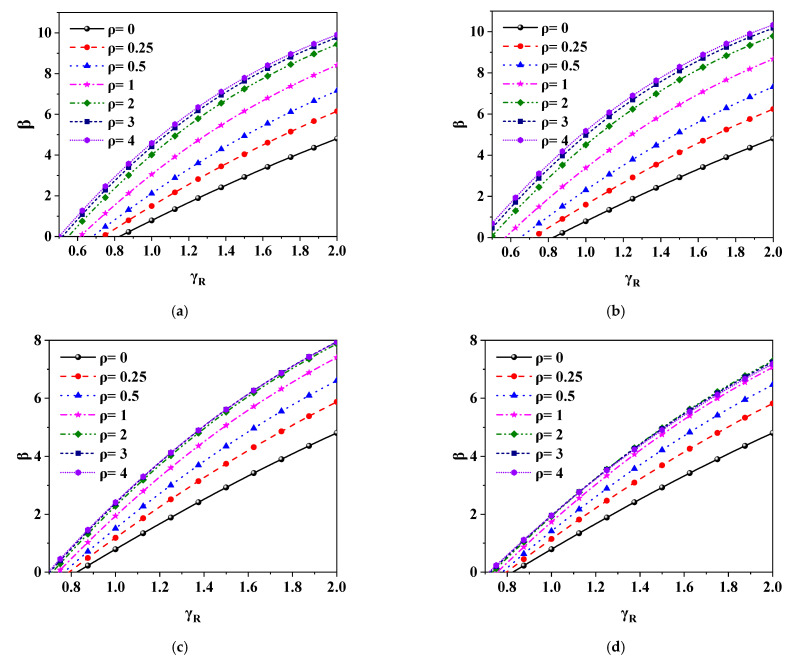
The relationship between *β* and *γ*_R_ under different load types and load ratios (grade II): (**a**) D + R; (**b**) D + O; (**c**) D + W; (**d**) D + S.

**Figure 8 polymers-14-01418-f008:**
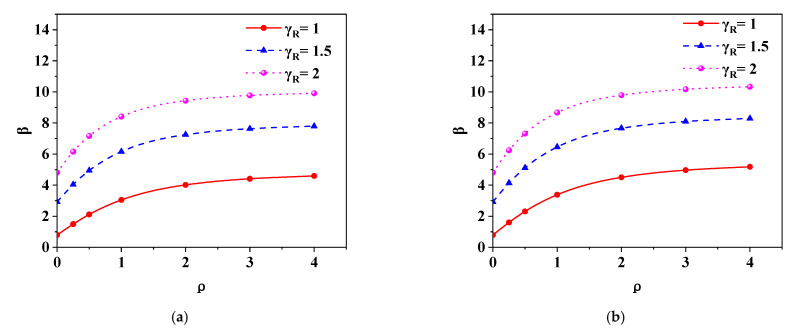
The relationship between *β* and *ρ* under different load types and partial factors for resistance (grade II): (**a**) D + R; (**b**) D + O; (**c**) D + W; (**d**) D + S.

**Table 1 polymers-14-01418-t001:** Classification results of *P. edulis* bamboo.

Grade		I	II	III
Grade boundary (GPa)		<16.5	(16.5, 18.5)	≥18.5
Quantity *n*		22	101	32
Proportion (%)		14.19	65.16	20.65
*MOE*	Mean	15.90	17.41	19.22
	5% quantile	15.25	16.57	18.56
	*CV*/%	2.35	3.12	2.59

**Table 2 polymers-14-01418-t002:** Bending strength statistics.

		Grade I			Grade II			Grade III		
		Normal	Lognormal	Weibull	Normal	Lognormal	Weibull	Normal	Lognormal	Weibull
*m*_f_/MPa	100%	130.94	130.95	130.66	131.84	131.85	131.45	136.46	136.47	136.20
	75%	128.44	128.45	128.54	129.14	129.14	129.10	133.58	133.59	133.60
	50%	125.89	125.90	125.99	126.72	126.72	126.70	130.87	130.87	130.88
	25%	122.62	122.62	122.46	123.83	123.83	123.79	127.56	127.56	127.66
*CV*	100%	0.051	0.051	0.060	0.048	0.047	0.060	0.051	0.051	0.059
	75%	0.040	0.040	0.037	0.034	0.034	0.037	0.039	0.039	0.039
	50%	0.035	0.036	0.033	0.025	0.025	0.027	0.031	0.031	0.031
	25%	0.026	0.026	0.030	0.013	0.013	0.014	0.021	0.021	0.016
*f*_k_/MPa	100%	120.83			122.73			126.04		
	75%	118.47			121.86			126.04		
	50%	118.47			121.47			122.02		
	25%	118.47			121.45			122.02		

**Table 3 polymers-14-01418-t003:** Partial factors for resistance under target reliability *γ*_Ri_.

	Load Combination	*γ* _Ri_						
		*ρ* = 0	*ρ* = 0.25	*ρ* = 0.5	*ρ* = 1.0	*ρ* = 2.0	*ρ* = 3.0	*ρ* = 4.0
Grade I	D + R	1.66	1.41	1.24	1.08	0.94	0.90	0.87
	D + O	1.66	1.38	1.22	1.03	0.88	0.83	0.80
	D + W	1.66	1.46	1.35	1.25	1.18	1.17	1.17
	D + S	1.66	1.47	1.37	1.28	1.26	1.26	1.26
Grade II	D + R	1.69	1.43	1.27	1.10	0.96	0.91	0.89
	D + O	1.69	1.41	1.24	1.05	0.90	0.84	0.82
	D + W	1.69	1.49	1.38	1.27	1.20	1.19	1.19
	D + S	1.69	1.50	1.40	1.31	1.28	1.28	1.28
Grade III	D + R	1.70	1.44	1.28	1.10	0.97	0.92	0.90
	D + O	1.70	1.41	1.25	1.05	0.90	0.85	0.82
	D + W	1.70	1.49	1.39	1.28	1.21	1.20	1.19
	D + S	1.70	1.50	1.40	1.32	1.29	1.29	1.29

**Table 4 polymers-14-01418-t004:** Bending strength design values of *P. edulis* bamboo of different grades.

	Grade I	Grade II	Grade III
*f*_d_ (MPa)	27.40	27.69	28.47

## Data Availability

Not applicable.

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
