# Peer review of "Bending Strength Design Method of Phyllostachys edulis Bamboo Based on Classification"

_polymers, 2022, doi:10.3390/polym14071418_

Round 1

Reviewer 1 Report

Reviewers’ comments

Manuscript number: polymers-1648498

Title: Bending strength design method of Phyllostachys Edulis bamboo based on classification.

Article Type: Article

Comments: 

The manuscript needs a detailed editing. Some markings are made to just illustrate the extent of editing needed. A thorough revision addressing all the concerns is needed and if the authors are prepared to do that it can be considered for a review of the revised manuscript.

- In the Abstract - the authors need to improve with more specific short results and conclusions, i.e. academic novelty or technical advantages.

- Line number 9 - Phyllostachys Edulis (P. edulis)…to…. Phyllostachys Edulis (P. edulis).

- Introduction part should be detailed, it is useful for new readers.

- 2.3 Reliability analysis - should be improve.

- Please provides the references for all equations and formula.

- The authors are obliged to repeat the results part of the 3.1. Destruction process and destruction form.

- 3.6. Determination of design values of bending strength - should be improve.

- Conclusion – too long, should be concise.

- Several faults: are added or missing spaces between words: see manuscript file.

- References: make all references in same format for volume number, page number and journal name, because it is difficult to searching and reading.

- Add the graphical abstract, it is use full to readers.

- Minor English corrections is required throughout the manuscript.

So that I recommended this manuscript to major revision and for future process.

Author Response

Thank you for your review comments! We have revised the paper according to your comments. Please refer to the attachment for details.

Reviewer 2 Report

This article selects Moso bamboo as the research object and measures the flexural modulus of elasticity of the samples through the four-point bending experiment. The Moso bamboo is divided into three categories according to the different values. The experimental method is set reasonably, and the results are discussed more thoroughly. However, there are the following problems:

As much as I prefer to go straight to the testing itself, in this paper, it will be better to have a bit more detail in the introduction with previous literature.

The test samples are all about 4-year-old Moso bamboo (lines 91-92), so the result obtained is the classification of Moso bamboo at this age, which is of little practical significance.

What is the max load of the testing machine?

Do you test modulus with a mechanical or video extensometer?

Figure 1. The scale bar within the photos should be reported.

Classification of Moso bamboo by bending test, which does not involve the field of polymers, and the article lacks novelty.

The research work is too little; only a few tests and very basic statistics have been applied. My opinion is, therefore, to add some more analysis to it.

According to a plagiarism report tested by Turnitin, around 7% was copied from “Pengcheng Liu, Qishi Zhou, Hai Zhang, Jiefu Tian. “Design strengths of bamboo-based on reliability analysis”, Wood Material Science & Engineering, 2021”; for example, check line 128-133.  I have added the Turnitin report anonymously.

Author Response

(The authors gave the same response as above.)

Reviewer 3 Report

      This work provides a analysis of the potential of Phyllostachys Edulis bamboo as a structural material. The bending strength test and classification of P. edulis bamboo were carried out, the factors affecting the reliability were analyzed, and the design values of the bending strength of P. edulis bamboo were proposed based on the reliability analysis. I think this work is foundational. Therefore, I recommend the journal accept this research after the authors have completed some necessary corrections. Some comments are listed below:

  1. Some statements are inappropriate and not academic enough. For example, on page 1, line 42, the statement "the design and construction of bamboo buildings are flexible" is inappropriate. You could say they are convenient. This is just one example, and there are many similar situations in the manuscript;
  2. The statement in Section 3.1 is not acceptable without data and evidence to support, and the author should provide the stress-strain curve for each specimen;
  3. There are many invalid and redundant sentences in the article, which makes the article too long. For example, lines 152 to 159 in Section 3.2;
  4. The number of references cited is too small, relative to the large volume;
  5. Most of the literature citations are concentrated in the INTRODUCTION and MATERIALS AND METHODS part. Some explanations lack the support from data or references. It is worrying about the accuracy of some conclusions drawn by the author;
  6. The statement "For ductile failure, the target reliability β0 is 3.7, and for brittle failure, the target reliability β0 is 3.7" at line 209 is confusing. Please author check here for errors;
  7. Statements on lines 212 to 215 are duplicated;
  8. If possible, please try to enlarge the words in all illustrations in the manuscript.

Author Response

(The authors gave the same response as above.)

Round 2

Reviewer 1 Report

Now the paper is suitable for publication in this journal. It can be accepted.

Reviewer 2 Report

It can be considered in present form.